# Analysis of the Electrical and Thermal Properties for Magnetic Fe_3_O_4_-Coated SiC-Filled Epoxy Composites

**DOI:** 10.3390/polym13183028

**Published:** 2021-09-07

**Authors:** Jiale Wu, Yiran Zhang, Yangzhi Gong, Kun Wang, Yun Chen, Xupeng Song, Jun Lin, Boyang Shen, Shaojian He, Xingming Bian

**Affiliations:** 1State Key Laboratory of Alternate Electrical Power System with Renewable Energy Sources, North China Electric Power University, Beijing 102206, China; wujiale2016@126.com (J.W.); gongyangzhi@163.com (Y.G.); songxupeng96@163.com (X.S.); Jun.lin@ncepu.edu.cn (J.L.); 2State Grid Maintenance Company Yichang Operation Maintenance Branch, Yichang 443000, China; yr495459435@163.com; 3State Key Laboratory of Advanced Power Transmission Technology, Global Energy Interconnection Research Institute, Co., Ltd., Beijing 102211, China; wangkun278@163.com (K.W.); yyaeyyae@163.com (Y.C.); 4Electrical Engineering Division, University of Cambridge, Cambridge CB3 0FA, UK; bs506@cam.ac.uk

**Keywords:** magnetization treatment, orderly arrangement, breakdown strength, dielectric properties, thermal conductivity, silicon carbide

## Abstract

Orderly arranged Silicon carbide (SiC)/epoxy (EP) composites were fabricated. SiC was made magnetically responsive by decorating the surface with iron oxide (Fe_3_O_4_) nanoparticles. Three treatment methods, including without magnetization, pre-magnetization and curing magnetization, were used to prepare SiC/EP composites with different filler distributions. Compared with unmodified SiC, magnetic SiC with core-shell structure was conducive to improve the breakdown strength of SiC/EP composites and the maximum enhancement rate was 20.86%. Among the three treatment methods, SiC/EP composites prepared in the curing-magnetization case had better comprehensive properties. Under the action of magnetic field, magnetic SiC were orderly oriented along the direction of an external field, thereby forming SiC chains. The magnetic alignment of SiC restricted the movement of EP macromolecules or polar groups to some extent, resulting in the decrease in the dielectric constant and dielectric loss. The SiC chains are equivalent to heat flow channels, which can improve the heat transfer efficiency, and the maximum improvement rate was 23.6%. The results prove that the orderly arrangement of SiC had a favorable effect on dielectric properties and thermal conductivity of SiC/EP composites. For future applications, the orderly arranged SiC/EP composites have potential for fabricating insulation materials in the power electronic device packaging field.

## 1. Introduction

With the development of UHV transmission technology, the reliability requirements for the electrical equipment in AC/DC transmission systems with different voltage levels have become more and more strict [1,2,3]. High-voltage high-power electronic devices (such as IGBT and IGCT) are the core components for manufacturing various high-voltage large-capacity power converters and control equipment [4]. As a kind of stable electrical insulation material with good processing performance, corrosion resistance and bonding properties, epoxy resin is widely used in the electronic packaging field [5,6]. The electronic devices packaged by epoxy resin have excellent integrity and dimensional stability, which can effectively prolong the service life of electronic chips by 2–3 times [7]. However, the low initial thermal conductivity of epoxy resin limits its application in the thermal management of high-power electronic devices [8]. In practical applications, the heat generated by these high-power density devices causes the temperature rise, which has a negative impact on the insulation performance of the packaging materials, thus shortening the service life of the devices [9,10]. For example, the failure rate of the electronic equipment increases by 100% when the operating temperature increases by 10 °C [11]. Therefore, the development trend of packaging materials for high-voltage high-power electronic devices is to fabricate composite materials with good thermal conductivity and electrical insulation properties [12,13].

For the sake of solving the problems above, scholars introduced inorganic fillers with high thermal conductivity and good insulating properties into EP matrix to improve the electrothermal performance of the composites [14,15,16,17,18,19,20]. Wu et al [21] introduced reticulated porous alumina ceramics (RPCs) into EP doped with the tetrapod zinc oxide whiskers (T-ZnOw) and EP composites with a continuous network structure were prepared. Compared with the pure EP, the thermal expansion coefficient of the EP composites was reduced by one order of magnitude and the elastic modulus was increased by 5.7 times. Takahiro et al [22] filled EP matrix with the nano-layered silicate and micro silica (SiO_2_). Because the nano-silicate was wrapped around the SiO_2_, the EP composites had a more compact internal structure and the breakdown strength was raised by 7%, compared with that of EP composites filled only with SiO_2_. Stukhlyak P.D. et al [23] explored the shock-fracture mechanism of different scales by taking SiC/epoxy composites with different filling content as the research object. The results show that the high heterogeneity degree of epoxy composites at different structural scales is one of the factors that affect their physical and mechanical properties. Buketov A. et al [24] investigated the effects of the nanofiller content on the structural transformation process of epoxy composites during cross-linking and thermal destruction by taking carbon nanotubes/epoxy composite materials as the research object. The optimal filling content of the carbon nanotubes in this composite system was determined. Aphesteguy, J.C. et al [25] fabricated two series of Fe_3_O_4_ and Zn_x_Fe_3−x_O_4_ NPs with different compositions by a co-precipitation method under different conditions and studied the effects of Zn concentration on the structural, magnetic and microwave properties of the prepared material. Yuvchenko, A.A. et al [26] studied the magnetic impedance (MI) of film elements in the form of meanders with a layered structure and variable geometry and determined the effects of stray magnetic fields generated by spherical iron particles or variously configured ferrofluids containing iron oxide nanoparticles.

The direct method to improve the thermal conductivity of EP composites was to establish the heat transfer path in the polymer matrix [27,28]. Recently, many researchers have turned their attention to the construction of an ordered network of fillers. Under the action of an external electromagnetic field, the filler particles can achieve directional arrangement in the EP matrix with the help of functional nanoparticles attached on the surface. EP composites with an excellent comprehensive performance can be prepared by forming an ordered filler network [29,30,31,32,33]. Wu et al [34] used an AC electric field to regulate the graphene nanosheets (GnPs) filled in an EP matrix. The key parameters that controlled the deflection and chain formation of the GnPs were determined by theoretical models. The results suggest that the prepared EP composites had significantly improved electrical and thermal conductivity in the orderly arrangement direction of the GnPs fillers. When the GnPs were oriented perpendicular to the crack-development direction, the fracture toughness was greatly enhanced. In [35], through in situ electrophoretic force, barium titanate (BaTiO_3_) was driven to form a chain-like arrangement in a high electric field strength area and remained random in a low electric field strength area. It was found that the insulation performance of the prepared EP composites with a gradient dielectric constant was improved, the partial discharge reduced significantly and the AC flashover voltage increased by 31.8%. Zheng et al prepared the magnetic three-phase composite material of EP, aluminum nitride (AlN) and nickel (Ni). The results show that the anisotropic thermal conductivity of the prepared composites was obtained with the chain structure under external magnetic field [36]. Kurlyandskaya, G.V. et al [37] took magnetic composites with nickel nanoparticles synthesized by the method of the electrical explosion of wire as the research object and explored the influences of Ni content and MNPs aggregation degree on magnetic and microwave properties of Ni nanoparticles–acrylic copolymer composites.

Silicon carbide (SiC) has the advantages of moderate costs, low chemical activity and excellent thermal conductivity, which make it a strong candidate among variety of inorganic fillers. In this paper, nano-sized Fe_3_O_4_ is coated on the surface of SiC fillers by chemical deposition to make SiC ferromagnetic. During the preparation of the SiC/EP composites, the magnetic SiC is regulated by three treatment methods, including without magnetization, pre-magnetization and curing magnetization. The effects of SiC distribution on breakdown strength, dielectric properties and thermal conductivity of SiC/EP composites are explored. It is expected that this research can provide experimental basis for the magnetization treatment of inorganic fillers, further expand possible applications of EP composites and meet the high requirements of choosing appropriate insulating materials for the electronic chip internal packaging.

## 2. Materials and Methods

Two kinds of SiC fillers were supplied by Qinhuangdao Yinuo Material Co., Ltd., (Qinhuangdao; China). The material parameters of the SiC fillers are listed in Table 1. Epoxy resin (E-51) was provided by Shanghai Resin Factory Co., Ltd., (Shanghai, China). Methylcyclohexene-1,2-dicarboxylic anhydride (C_9_H_10_O_3_) and 2,4,6-Tris (dimethylaminomethyl) phenol (C_15_H_27_N_3_O), used as curing agent and accelerator, respectively, were provided by TCI Development Co., Ltd., (Shanghai, China). Sodium hydroxide (NaOH), ferrous sulfate heptahydrate (FeSO_4_·7H_2_O), ferric chloride hexahydrate (FeCl_3_·6H_2_O), hydrochloric acid (HCl), absolute ethanol and sodium polystyrene sulfonate (PSS), used during the synthesis process of the magnetic SiC fillers, were provided by Beijing Innochem Technology Co. Ltd., (Beijing, China).

### 2.1. Sample Preparation

#### 2.1.1. Synthesis of Magnetic SiC

SiCp and SiCw (collectively referred to as SiC in the following description) were pre-modified with PSS aqueous solution respectively to make the SiC surface attach negative ions, so that the SiC surface had negative charge and was easy in absorbing and combining positive ions. The PSS pre-modified SiC was fully mixed with a solution containing Fe^2+^ and Fe^3+^ ions, then alkaline titration was carried out. In theory, a layer of magnetic Fe_3_O_4_ nanoparticles would be coated on the surface of the pre-modified SiC by chemical deposition [38,39]. The reaction diagram is illustrated in Figure 1.

(A)PSS pre-modified SiC(a)20 g of SiC powder was added to 120 mL of absolute ethanol and 80 mL of ultrapure water, mixed thoroughly and dispersed with an ultrasonic disperser for 50 min to obtain the SiC suspension.(b)An appropriate amount of PSS powder was weighed and added into the SiC suspension, in which the content of PSS was 0.1 mol/L. The mixed solution was stirred at a uniform speed of 300 r/min for 30 min.(c)HCl was added into the well mixed solution; we detected the pH value of the solution until pH = 3. The obtained solution was heated to 30 °C in an oil bath and stirred at a uniform speed of 300 r/min for 2 h.(d)After the solid–liquid separation of the mixed solution, the powder was washed to neutral with absolute ethanol and dried in vacuum to obtain PSS-modified SiC, which was denoted as SiC-PSS.

(B)Further modified SiC-PSS

In this part, the SiC-PSS was further modified and a layer of nano-magnetic-Fe_3_O_4_ was coated on the surface of the pre-modified SiC by the chemical deposition method to generate magnetic SiC. The chemical reaction formula is as follows.
Fe^3+^ + Fe^2+^ + OH^−^ → Fe_3_O_4_ + H_2_O

The preparation process of the magnetic SiC (M-SiC) is illustrated in Figure 2.

(a)3 g of SiC-PSS was added to 200 mL of absolute ethanol and 100 mL of ultrapure water, mixed thoroughly and dispersed with an ultrasonic disperser for 1 h to obtain the SiC-PSS suspension.(b)4.17 g of FeSO_4_·7H_2_O and 0.811 g of FeCl_3_·6H_2_O (mole ratio is 5:1) were weighed and dissolved in 300 mL of ultrapure water to prepare an Fe^2+^ and Fe^3+^ mixed solution. The solution was stirred until the powder was completely dissolved and the color of the solution was pale yellow and transparent.(c)The SiC-PSS suspension and the prepared Fe^2+^ and Fe^3+^ solution were mixed in a round-bottom flask. The mixed solution was heated to 30 °C in the oil bath and stirred at a uniform speed of 300 r/min for 90 min in the absence of air.(d)300 mL of NaOH solution with a concentration of 0.5 mol/L was prepared, the stirring speed was set to 200 r/min and the prepared NaOH solution was added dropwise until the pH value of the solution reached 12–13. After stirring at low speed for 5 min, the temperature of oil bath was raised to 50 °C, then the solution was crystallized at a constant temperature for 1.5 h.(e)After the solid–liquid separation of the mixed solution, the powder was washed to neutral with absolute ethanol and dried in vacuum to obtain magnetic SiC, which was denoted as M-SiC.

#### 2.1.2. Fabrication of the Aligned SiC/epoxy Composites

The Fe_3_O_4_ nanoparticle has good magnetic response properties. In the experiment, Fe_3_O_4_ could be coated on the surface of SiC by the wet chemical co-deposition method, which offered SiC with considerable ferromagnetism. The self-assembly of magnetic particles under the magnetic field comes from two aspects. Firstly, the particles have magnetic anisotropy; after being subjected to the magnetic field force, a magnetic moment is generated between the axis of the easy magnetization direction and the magnetic field, so that the easy magnetization axis of the particles is aligned along the direction of the magnetic force line. Secondly, the particles produce spin magnetization, so that each magnetic particle becomes a magnetic dipole. There is a magnetic coupling moment between the magnetic dipoles and the mutual repulsion or attraction between the magnetic coupling moments, which promotes the orderly arrangement of the magnetic particles. M-SiC has a certain magnetic response ability. Under the action of the external magnetic field, the M-SiC fillers would be arranged in a one-dimensional chain-like ordered structure along the magnetic field line [40,41,42], as shown in Figure 3.

According to the difference of filler type and magnetic field action mode, the composites prepared in the experiment were divided into 4 types. The first one was composites filled with raw SiC (R-SiC) without modification. The second one was composites filled with M-SiC fillers but with no external magnetic field applied during preparation, namely, without magnetization. The third one was composites filled with M-SiC fillers and the external magnetic field was applied before the curing process, namely, pre-magnetization. The last one was composites filled with M-SiC fillers and the external magnetic field was applied during the entire curing process, namely, curing magnetization. The process is shown in Figure 4.

(a)The epoxy resin was poured into a three-necked flask, heated to 60 °C in an oil bath to improve the fluidity of the matrix and mechanically stirred for 30 min to discharge the water vapor adsorbed by the epoxy during storage.(b)An appropriate amount of curing agent and M-SiC (or R-SiC) was weighed and added into the epoxy resin. The suspension was heated to 60 °C in an oil bath and stirred at a uniform speed of 360 r/min for 1 h.(c)After the epoxy matrix and SiC filler were fully mixed, the suspension was treated with an ultrasonic disperser for 20 min. An appropriate amount of accelerator was added and the mixture was stirred at a constant speed of 260 r/min for 10 min.(d)The mixed solution was taken out and it was placed in a vacuum-drying oven. The vacuum operation was carried out at 60 °C until no obvious bubbles overflowed. The molds were sprayed with release agent, then put into the blast dryer for preheating.

The preparation steps of the 4 types of materials were the same in the first four steps and the next step was different according to the material.

(e)For the first and second type of composite material, the vacuum-degassed solution was poured into the mold and cured according to the heating curve of 100 °C for 4 h and 150 °C for 10 h. The samples were taken out after natural convection cooling. Since there was no magnetic field applied during the curing process, the first type of composites obtained was denoted as R-SiC/EP-N and the second type was denoted as M-SiC/EP-N. For the third type of composite material, the mold containing the vacuum-degassed solution was required to be placed in a constant magnetic field for a period of time (0.2 T, 20 min); then, the sample was cured according to the heating curve after the magnetic field was removed. This type of method was pre-magnetization and the obtained composites were denoted as M-SiC/EP-P. The fourth type of composite was placed in a constant magnetic field (0.1 T) produced by a permanent magnet during the whole curing process. This type of method was called curing magnetization and the obtained composites were denoted as M-SiC/EP-C. Different types of composites prepared in the experiment and their abbreviations are shown in Table 2.

### 2.2. Measurements

A scanning electron microscope (SEM) (Quanta FEG 250, FEI, Hillsboro, OR, USA) was used to inspect the morphologies of SiC powder before and after Fe_3_O_4_-coating modification and to investigate the cross-sections of the SiC/epoxy composites. The samples were sputtered with a thin layer of gold before SEM observations to avoid charge accumulation. The energy-dispersive X-ray spectroscopy (EDX) was performed using the EDX attachment of the SEM machine to analyze the elements contained in the magnetic SiC powder.

Infrared spectra were obtained from a FTIR spectrometer (Perkin Elmer Frontier, Waltham, MA, USA) with an accumulation of 100 scans at a resolution of 2 cm^−1^.

Magnetic properties of raw and magnetic SiC powders were measured on a vibrating scanning magnetometer (VSM, Lake shore, 7404 series VSM).

Breakdown strength was measured with a dielectric strength tester (HCDJC-50kV, Beijing Huace Testing Instrument Co. Ltd., Beijing, China) at ambient temperature with an increasing alternating voltage (50 Hz) of 2 kV/s. The samples sandwiched between two copper rod electrodes with a diameter of 25 mm were immersed in pure silicone oil to prevent surface flashover.

Dielectric properties were measured at ambient temperature using a Novocontrol GmbH concept 80 Broadband Dielectric Spectrometer over a frequency range of 10–10^6^ Hz. A sample with a size of 40 mm diameter and 1mm thickness was placed in the middle of the fixture. The change rule of the complex permittivity with frequency was obtained, which reflected the internal polarization and lattice vibration of the sample.

Based on the transient plane heat source method, the thermal conductivity of the prepared composites was measured by the thermal conductivity meter (Hot Disk TPS 2500S) at 25 °C with reference to the international standard ISO 22007-2. During the test, the probe was clamped with two samples with a size of 70 mm diameter and 3 mm thickness. Since the heat conduction process inside the material was complicated, it is assumed that the heat only flowed in the material during the heating process. The heat transfer depended on the temperature change of the probe.

## 3. Results and Discussion

### 3.1. Characteristics

The morphology of SiC before (R-SiC) and after (M-SiC) chemical deposition was observed by SEM. Figure 5a,b shows the SEM images of SiCp and SiCw before deposition. The raw SiCp had a relatively regular polyhedral structure and the raw SiCw was columnar. The surface of raw SiC was relatively smooth. Figure 5c,d shows the SEM images of SiCp and SiCw after deposition. It can be clearly seen that the surface roughness of M-SiCp and M-SiCw increased. The particle size information in Figure 5c,d indicates that the nanoparticles were successfully coated on the surface of M-SiCp and M-SiCw by the chemical deposition method (as shown in the red circle). The EDX spectra of M-SiC is shown in Figure 5e. The abscissa is X-ray energy (keV) and the type of the element can be determined according to the position of the spectral peak. The ordinate cps/eV (counts per second per eV) is the signal intensity, which corresponds to the element content. The EDX spectrum semi-quantitatively tested the elements contained in M-SiC powder. It can be seen that, in addition to the Si and C elements contained in SiC itself, the peaks corresponding to the Fe and O elements also appeared, which proves that Fe and O elements attached on the surface of M-SiC.

The FTIR spectra of R-SiC and M-SiC are shown in Figure 6a,b. In the FTIR spectra of R-SiC, there is a characteristic absorption peak near wavenumber 800 cm^−1^, which corresponds to the in-plane stretching vibration peak of Si-C. In the FTIR spectra of M-SiC, the characteristic peak of Si-C also appears near 800 cm^−1^, which proves that the basic structure of M-SiC did not change after modification. Meanwhile, the characteristic peak of 554 cm^−1^ appears on the FTIR spectra of M-SiC, which corresponds to the stretching vibration peak of Fe-O [43], meaning that the nanoparticles deposited on the surface of M-SiC were composed of iron atoms and oxygen atoms.

Magnetic hysteresis loops of M-SiC are shown in Figure 6c, the saturation magnetization of surface-modified M-SiC was about 24 emu/g. The macro magnetic response ability of M-SiC was also tested. M-SiC powder was dispersed thoroughly in absolute ethanol. The mixed solution was placed next to a permanent magnet with a magnetic field strength of about 0.05 T and the movement of M-SiC over time *t* was observed, as shown in Figure 6d–g. It can be clearly seen that, under the action of magnetic field, M-SiC moved to the side of permanent magnet in a short period of time and, finally, accumulated around the magnet. The mixed solution changed from the opaque brown state to the solid–liquid separation state, which indicated ferromagnetic particles were attached to the surface of M-SiC after the modification. M-SiC fillers can move directionally in the magnetic field and have good magnetic responsiveness. This result is consistent with the results of SEM, EDX, FTIR and VSM, meaning that the magnetic Fe_3_O_4_ nanoparticles deposited on the surface of M-SiC.

The SEM images of the randomly and orderly oriented SiC/EP composites are shown in Figure 7. As shown in Figure 7a, when there was no external magnetic field, the M-SiC fillers were randomly dispersed in the epoxy matrix and did not make physical contact with each other. In Figure 7b, the M-SiC fillers were oriented along the direction of the magnetic field under the action of an external field and made physical contact with each other, forming a chain.

### 3.2. Breakdown Strength

Breakdown strength is an important parameter to estimate the electrical characteristics of EP composites. The results of breakdown experiments are usually processed by the two-parameter Weibull distribution function and the formula is as follows.
(1)$$F\left(E_{b};\;\alpha ,\;\beta \right)=1-\exp(-{\left(\frac{E_{b}}{\alpha }\right)}^{\beta })$$
where *E*_b_ is the test value of the breakdown strength (the unit is kV/mm); *α* is the scale factor which represents the breakdown strength when the breakdown probability is 63.2% (the unit is kV/mm) and the larger the value of *α*, the higher the breakdown strength is; *β* is the shape factor, which is the slope of the fitted curve, and the larger the value of *β*, the more uniform the breakdown data are. The breakdown probability distribution function *F* (*i*, *n*) of Ross function is as follows.
(2)$$F\left(i,n\right)\approx \frac{i-0.44}{n+0.25}\times 100\%$$
where *n* is the total number of samples, *n* = 10, and *i* is the sample number.

Weibull distributions of breakdown strength of the SiC/EP composites are shown in Figure 8. In Figure 8a, the composites filled with 5 wt% R-SiCp and without magnetization treatment had a breakdown strength of 22.4 kV/mm. With the increase in R-SiCw filling content, the breakdown strength of the R-SiC/EP-N composites rose first, then decreased. The value reached the relative maximum at R-SiC_p5w0.5_/EP-N, which was 31.2 kV/mm. In Figure 8b–d, for the three cases of without magnetization, pre-magnetization and curing magnetization, the change trends of breakdown strength of the M-SiC/EP composites were basically the same. All breakdown strengths of M-SiC/EP reached the relative maximum at M-SiC_p5w0.5_/EP, which were 35.2 kV/mm, 31.8 kV/mm and 31.2 kV/mm, respectively. Since the shape of the SiCw filler was columnar, if the direction of SiCw was perpendicular to the that of the electric field, it had a greater scattering effect on electrons. When the electrical treeing developed to the plane where the SiCw filler was located, due to its high critical breakdown strength, the electrical treeing was not able to directly breakdown the SiCw and could only develop along the interface between SiCw and matrix. The development path of electrical treeing was equivalent to bypassing the SiCw filler and extending to another electrode [44]. In this case, the growth of electrical treeing was blocked and the breakdown path grew. The addition of a small amount of SiCw was helpful to improve the breakdown performance of the composites. As the filling content of SiCw further increased, the introduced voids and defects increased. Although the breakdown path was longer at this time, the breakdown strength of the composites reduced due to the existence of excessive defects and voids.

It was found that the breakdown strength of R-SiC/EP-N was lower than that of M-SiC/EP-N under the same conditions. For example, the breakdown strength of R-SiC_p5_/EP-N was 22.4 kV/mm, while that of the M-SiC_p5_/EP-N composites was 25.0 kV/mm. By employing surface modification, the surface of M-SiC was coated with a layer of Fe_3_O_4_ nanoparticles, which was equivalent to the introduction of a core-shell structure. The existence of a nanoshell significantly reduced the distortion of electric field caused by filling a single type of fillers with high dielectric constant [45]. Thus, the breakdown strength of the M-SiC/EP-N composites with a core-shell structure had a certain degree of improvement, compared to R-SiC/EP-N, and the maximum enhancement rate was 20.86%. Among the three types of M-SiC/EP composites, M-SiC/EP-N had the highest breakdown strength. This result indicates that the orderly arrangement of the M-SiC filler had an adverse effect on the breakdown characteristics of the EP composite material to a certain extent. Due to the high conductivity of M-SiC, the current path was established by the ordered alignment of the M-SiC fillers under the action of the external magnetic field, which led to the decrease in the breakdown strength of the M-SiC/EP-P and M-SiC/EP-C composites. Compared to the M-SiC/EP-N, the decrease was about 10%, while, compared to the R-SiC/EP-N, there was almost no significant decrease.

### 3.3. Dielectric Properties

The dielectric constant spectra of composites with same components and different types are shown in Figure 9a–d. As the frequency increased, the relative dielectric constant *ε*_r_ of the composites gradually decreased. Due to the difference in electrical properties (dielectric constant and conductivity) between the matrix and SiC filler, free charge accumulation occurred at the interface of the two phases under the action of an external electric field, which resulted in the interface polarization [46]. With the increase in frequency, the delay time of the interface polarization increased and the relaxation phenomenon occurred. The polarization effect was weakened and, as a result, the dielectric constant gradually reduced. When no magnetic field was applied, the dielectric constant of the M-SiC/EP-N composites increased, compared to that of the R-SiC/EP-N composites filled with raw SiC. This was mainly due to the introduction of a nano-coating layer with a high dielectric constant on the surface of the SiC fillers during the preparation of M-SiC [47]. Among the three kinds of M-SiC/EP composites, M-SiC/EP-C had the lowest dielectric constant. As the action mode of magnetic field changed from without magnetization to curing magnetization, the distribution of the fillers in the composite material also changed from the random distribution to the ordered arrangement. As illustrated in Figure 10, when there was no external magnetic field, the M-SiC fillers inside the M-SiC/EP-N material were arranged disorderly and the movement of the epoxy macromolecular chain or the polar groups was almost unhindered. For the M-SiC/EP-C composites with curing magnetization, the M-SiC fillers had a clear orientation under the action of a magnetic field during the preparation process. The M-SiC fillers made physical contact with each other and arranged in an ordered manner, which might have limited the scope of movement of the epoxy molecular chain or the polar groups to a certain extent and lead to a decrease in the dielectric constant of the M-SiC/EP-C composites [48].

The dielectric constant spectra of the M-SiC/EP-C composites with different components are illustrated in Figure 9e. As the filling content of M-SiCw rose, the dielectric constant of the M-SiC/EP-C composites increased first, then reduced. On one hand, the addition of M-SiCw was equivalent to introducing a large number of new interfaces inside the composites. Under the action of the external electric field, due to the difference in electrical properties between M-SiCw and epoxy matrix, the charge accumulated on the new interfaces during the redistribution of internal voltage inside the M-SiC/EP-C composites, which aggravated the interface polarization to a certain extent and led to the increase in the dielectric constant of the material [46]. On the other hand, M-SiCw had ferromagnetism and could be oriented under the action of an external magnetic field to form an ordered alignment structure together with the M-SiCp filler during the curing magnetization process (as illustrated in Figure 10b). The ordered alignment structure might have restricted the movement of the epoxy macromolecular chain or the polar group to a certain extent and decreased the dielectric constant of the M-SiC/EP-C material [48]. The effect of the addition of M-SiCw on the dielectric constant of M-SiC/EP-C was a macroscopic reflection of the competitive relationship between the enhancement degree of interfacial polarization and the limitation degree of the epoxy macromolecular chain movement inside the EP matrix. When 0.1 wt% M-SiCw was added, the ordered alignment of M-SiC was not perfect due to the low filling content and large dispersion between the M-SiC fillers. The effect of the new interface introduced by M-SiCw on the interfacial polarization was greater than that on the movement of epoxy molecular chains or polar groups. As a result, the dielectric constant of the M-SiC_p5w0.1_/EP-C composites increased. With the further increase in the filling content of M-SiCw, more and new interfaces were introduced inside the composites and the contact probability of the M-SiC fillers was also increased at the same time. The ordered alignment structure formed by the M-SiC fillers inside the composites became more complete in the curing magnetization process. The limitation of M-SiCw on the movement of epoxy macromolecular chains or the polar groups was greater than the enhancement effect of the M-SiCw fillers on the interface polarization. Thus, the dielectric constant of M-SiC_p5w0.5_/EP-C, as well as that of M-SiC_p5w0.9_/EP-C, decreased.

Due to the introduction of the nano-coating layers with high dielectric loss on the surface of the R-SiC fillers during the preparation of M-SiC, the dielectric loss of M-SiC/EP-N composites increased, compared to that of R-SiC/EP-N composites filled with raw SiC. The dielectric loss spectra of the M-SiC/EP composite materials under the three action modes of the magnetic field are shown in Figure 11. As the frequency went up, the dielectric loss of the composites gradually increased. When the filling composition was the same, the dielectric loss of the M-SiC/EP-C composites was the lowest among the three, followed by M-SiC/EP-P, then by M-SiC/EP-N. Since the M-SiC fillers were oriented under the action of external magnetic field in the curing magnetization process, the formation of the ordered alignment structure in the M-SiC/EP-C composites limited the movement of polar groups or the macromolecular chains of epoxy [48], as illustrated in Figure 10b. The macroscopic performance was that the dielectric loss of the M-SiC/EP-C composites decreased. Comparing the spectra of M-SiC/EP-C in Figure 11a,b, it is found that the dielectric loss of the M-SiC/EP composites rose slightly when the filling content of M-SiCw increased from 0.1 wt% to 0.5 wt%. As the filling content of M-SiCw increased, the ordered alignment formed by M-SiC inside the material became more complete in the curing magnetization process. As a result, the carriers had a larger mean free path and could move in each semi-period of the alternating field, thus consuming more electric energy [49]. However, the dielectric loss tangent value of the M-SiC/EP-C composites was not large in the whole frequency range, which was about 0.012 and lower than that of M-SiC/EP-N. It is proved that orderly arrangement of M-SiC by applying the magnetic field during curing process was beneficial to reduce the dielectric loss of the M-SiC/EP composites.

### 3.4. Thermal Conductivity

The comparison of thermal conductivity of four kinds of SiC/EP composites is illustrated in Figure 12. Under the same conditions, R-SiC/EP-N showed a slightly higher thermal conductivity compared to the M-SiC/EP-N composites. In the preparation of M-SiC, the surface of M-SiC was coated with a layer of Fe_3_O_4_ nanoparticles, which was equivalent to the introduction of new interfaces, such as the SiC–Fe_3_O_4_ interfaces. Due to the differences in electronic and vibrational properties between SiC and Fe_3_O_4_ nanoparticles, when the energy carriers tried to traverse the interface, it first scattered at the interface. These interfacial phonon scattering phenomena lead to phonon interface resistance, which degraded the heat transfer performance [50].

Among the three types of M-SiC/EP composite materials, M-SiC/EP-C had the largest thermal conductivity, followed by M-SiC/EP-P, then by M-SiC/EP-N. For M-SiC/EP-N, as illustrated in Figure 13a, since no magnetic field was applied during the preparation process, the M-SiC fillers were randomly distributed inside the material and the heat conduction paths formed by the M-SiC fillers were entangled with each other. Therefore, the heat flux randomly transferred to all the directions, which resulted in low heat conduction efficiency inside the M-SiC/EP-N composites [51]. For M-SiC/EP-P, the original mixture of epoxy matrix and M-SiC was liquid and had strong fluidity. In the pre-magnetization stage, before curing, the M-SiC fillers were temporarily arranged in an orderly manner in the epoxy matrix due to the existence of an external magnetic field. When the external magnetic field was removed in the curing stage, the fluidity of the liquid and the forces on the M-SiC made the orderly arranged fillers become scattered. At the same time, the thermal movement of molecules was enhanced during the heating and curing process, which accelerated the transition process of the M-SiC fillers from orderly arrangement to disorderly distribution to some extent. Therefore, the thermal conductivity of the M-SiC/EP-N and M-SiC/EP-P composite materials was not much different. For M-SiC/EP-C, due to the continuous application of the magnetic field, the orderly arrangement state of the M-SiC fillers in the matrix could be maintained in the curing stage. The directional heat-conduction path (as shown in Figure 13b) was formed in the M-SiC/EP-C composites, which improved the heat transfer efficiency to a certain extent. The M-SiC/EP-C composites prepared by curing magnetization had a relatively optimal thermal conductivity.

It is found that with the increase of M-SiCw filling content, the thermal conductivity of the M-SiC/EP composites first increased, then decreased, all reaching the maximum at M-SiC_p5w0.5_/EP. The thermal conductivities of M-SiC_p5w0.5_/EP were 0.2266 W/m·K (without magnetization), 0.2553 W/m·K (pre-magnetization) and 0.2801 W/m·K (curing magnetization), respectively. When 0.1 wt% M-SiCw was added, the ordered alignment in the composites was not perfect due to the low filling content and large dispersion of the M-SiC fillers and the effect of improving the thermal conductivity of the composite material was limited. As the filling content of M-SiCw increased, the cubic-like M-SiCp and the columnar M-SiCw made physical contact with each other to gradually form the continuous heat transfer path. Phonons could easily move between the M-SiC fillers along these directional channels and the thermal conductivity was improved. However, with the further increase in M-SiCw filling content, the number of heat transfer path in the material tended to be saturated. Phonon scattering occurred at a large number of nanointerfaces introduced by M-SiCw [50], which led to the increase in phonon interface resistance and was not conducive to heat transfer. The thermal conductivity of the M-SiC/EP composites decreased when 0.9 wt% M-SiCw was added. 

The results suggest that the orderly arrangement of the M-SiC fillers had favorable effect on the thermal conductivity of the SiC/EP composite material. In the current experiment, the maximum thermal conductivity improvement rate was 23.6%.

## 4. Conclusions

Surface modification was performed to make SiC fillers showing ferromagnetism. SiC/EP composites were fabricated using three treatment methods, including without magnetization, pre-magnetization and curing magnetization. The effects of SiC distribution on the breakdown strength, dielectric properties and thermal conductivity of SiC/EP composites were explored. 

(1)When there was no external magnetic field, compared with unmodified SiC, magnetic SiC with core-shell structure was conducive to improve the breakdown strength of SiC/EP composites and the maximum enhancement rate was 20.86%. Under the action of external magnetic field, magnetic SiC was oriented orderly, aligned with the magnetic line and generated a directional current path. Thus, the orderly arrangement of SiC fillers can reduce the breakdown strength of SiC/EP composites to a certain extent, but the decrease is not significant compared to the composites filled with randomly arranged SiC.(2)Due to the introduction of a Fe_3_O_4_ nanolayer with a high dielectric constant (dielectric loss) in the process of surface modification, compared to the composites filled with raw SiC, the dielectric constant (dielectric loss) of the composites filled with the magnetic SiC fillers increased when there was no external magnetic field. Under the action of a magnetic field, magnetic SiC was orderly arrayed along the magnetic line and restricted the movement of the macromolecules or polar groups of the epoxy matrix, resulting in the decrease in the dielectric constant (dielectric loss) of the SiC/EP composites.(3)Due to the introduction of new interfaces in the preparation of magnetic SiC, the phonon scattering at the interfaces limited the heat transfer. When there was no external magnetic field, the thermal conductivity of composites filled with magnetic SiC decreased, compared to that of composites filled with raw SiC. Under the action of an external magnetic field, the magnetic SiC fillers were orderly aligned along the magnetic line and generated the directional heat conduction channels, which was beneficial in improving the thermal conductivity of SiC/EP composites, and the maximum improvement rate was 23.6%.(4)The results prove that the orderly arrangement of SiC had a favorable effect on the dielectric properties and thermal conductivity of the SiC/EP composites. For future applications, the orderly arranged SiC/EP composites have potential for preparing insulation material in the field of power electronic device packaging.

## Figures and Tables

**Figure 1 polymers-13-03028-f001:**
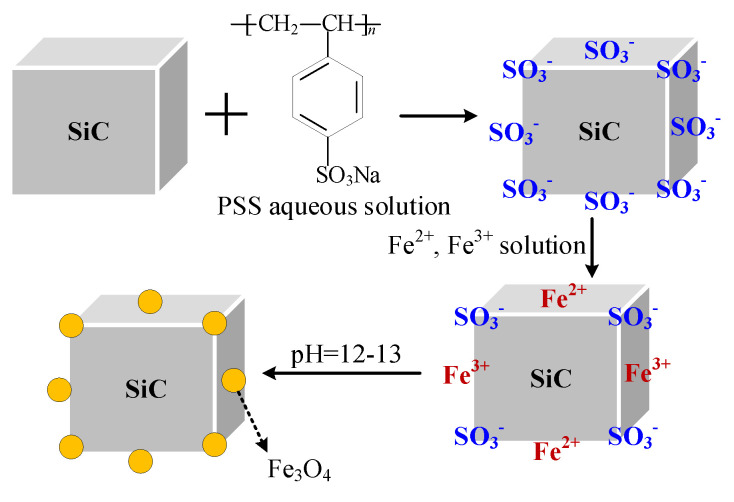
Synthesis process of Fe_3_O_4_-coated SiC.

**Figure 2 polymers-13-03028-f002:**
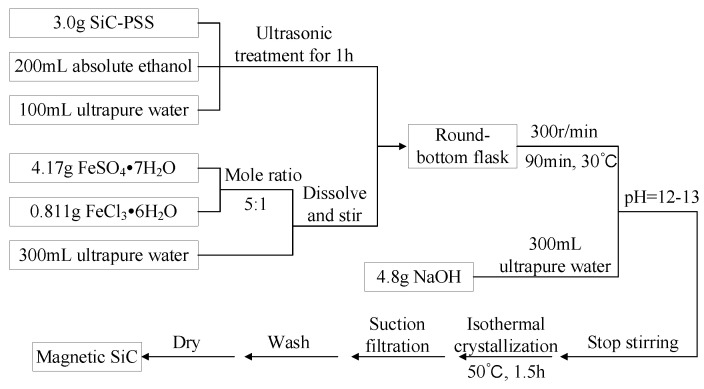
Preparation process of magnetic SiC (M-SiC).

**Figure 3 polymers-13-03028-f003:**
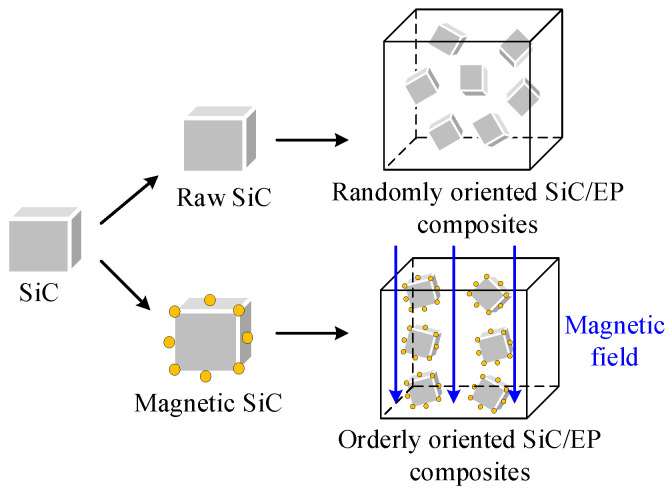
Schematic diagram of the randomly oriented SiC/EP composites and orderly oriented SiC/EP composites.

**Figure 4 polymers-13-03028-f004:**
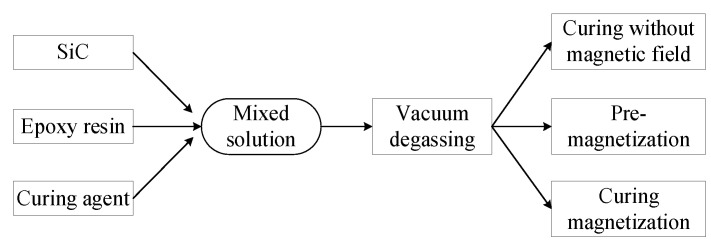
Preparation process of SiC/epoxy composites.

**Figure 5 polymers-13-03028-f005:**
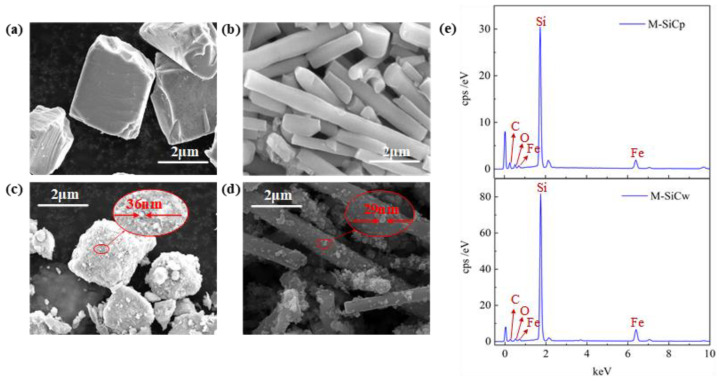
The SEM images and EDX spectrum of SiC before (R-SiC) and after (M-SiC) chemical deposition. (**a**) SEM image of R-SiCp; (**b**) SEM image of R-SiCw; (**c**) SEM image of M-SiCp; (**d**) SEM image of M-SiCw; (**e**) EDX spectra of M-SiC.

**Figure 6 polymers-13-03028-f006:**
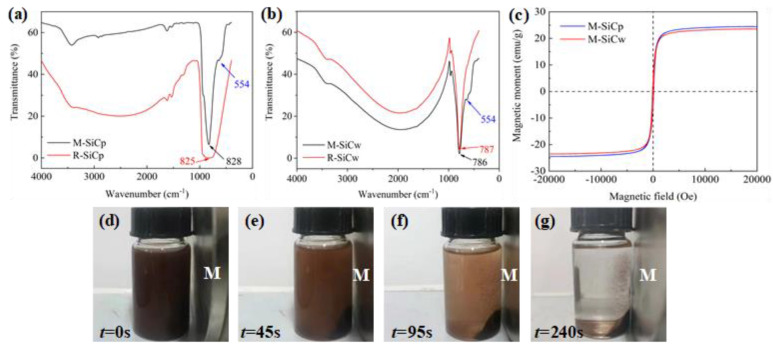
FTIR spectra and magnetic response phenomenon. (**a**) FTIR spectra of R-SiCp and M-SiCp; (**b**) FTIR spectra of R-SiCw and M-SiCw; (**c**) magnetic hysteresis loops of M-SiC; (**d**–**g**) magnetic response phenomenon of M-SiC in absolute ethanol (the letter M in the figure stands for permanent magnet).

**Figure 7 polymers-13-03028-f007:**
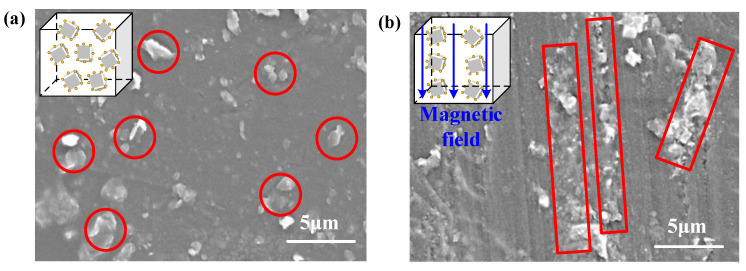
SEM images of randomly oriented SiC/EP composites and orderly oriented SiC/EP composites. (**a**) Randomly oriented M-SiC/EP-N composites; (**b**) orderly oriented M-SiC/EP-C composites.

**Figure 8 polymers-13-03028-f008:**
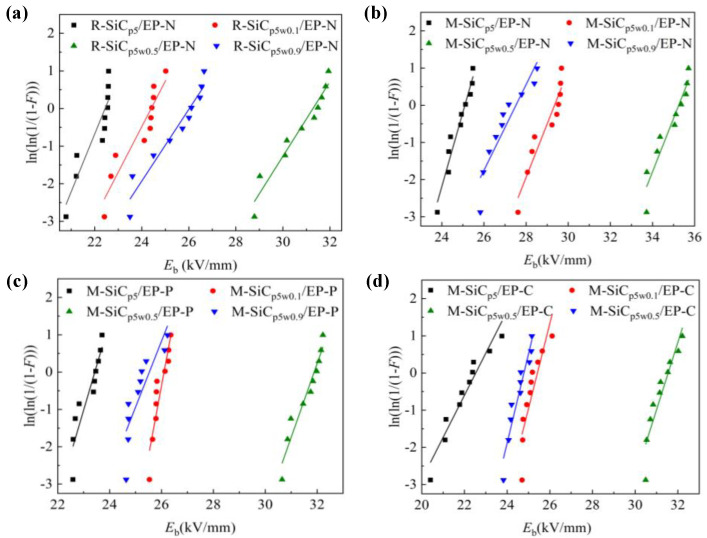
Weibull distribution of the breakdown strength of composites with same type and different components. (**a**) R-SiC/EP-N; (**b**) M-SiC/EP-N; (**c**) M-SiC/EP-P; (**d**) M-SiC/EP-C.

**Figure 9 polymers-13-03028-f009:**
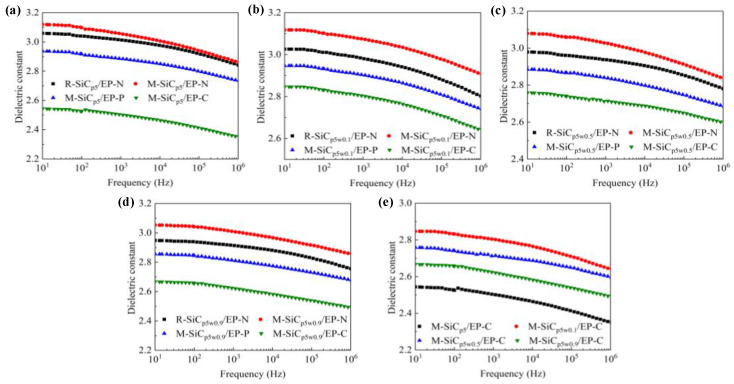
Dielectric constant spectra. (**a**) SiC_p5_/EP; (**b**) SiC_p5w0.1_/EP; (**c**) SiC/EP_p5w0.5_; (**d**) SiC_p5w0.9_/EP; (**e**) M-SiC/EP-C.

**Figure 10 polymers-13-03028-f010:**
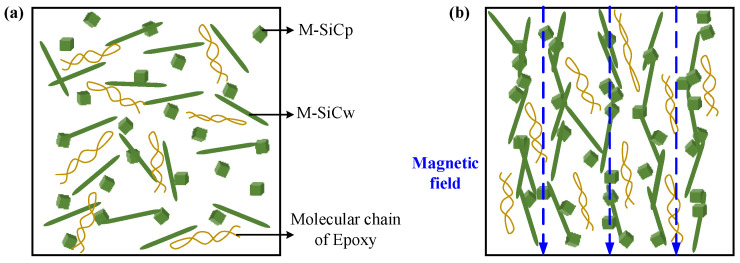
Effect of magnetic field action mode on the orientation of M-SiC filler and molecular chain inside M-SiC/EP composites. (**a**) Randomly oriented M-SiC/EP-N; (**b**) orderly oriented M-SiC/EP-C.

**Figure 11 polymers-13-03028-f011:**
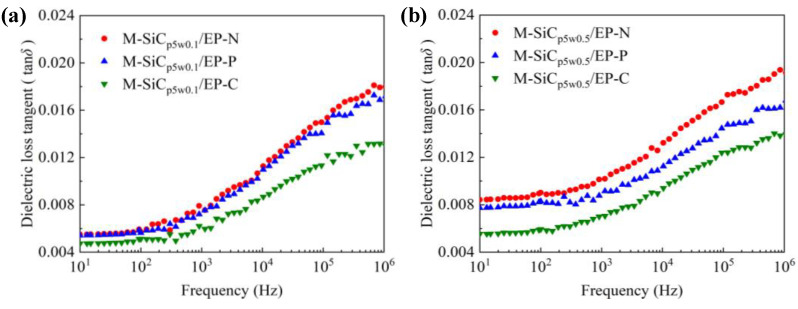
Dielectric loss spectra of M-SiC/EP composites under three action modes of magnetic field. (**a**) M-SiC_p5w0.1_/EP; (**b**) M-SiC_p5w0.5_/EP.

**Figure 12 polymers-13-03028-f012:**
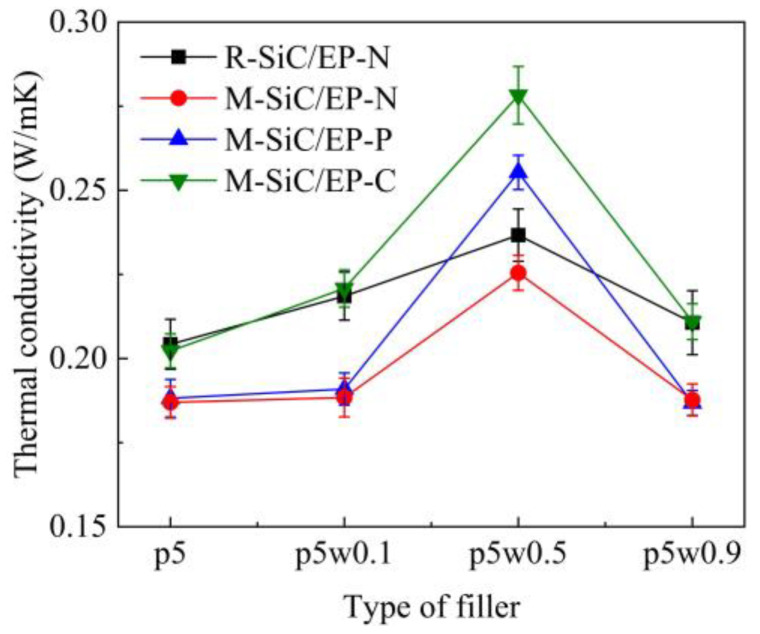
Comparison of thermal conductivity of four kinds of SiC/EP composites.

**Figure 13 polymers-13-03028-f013:**
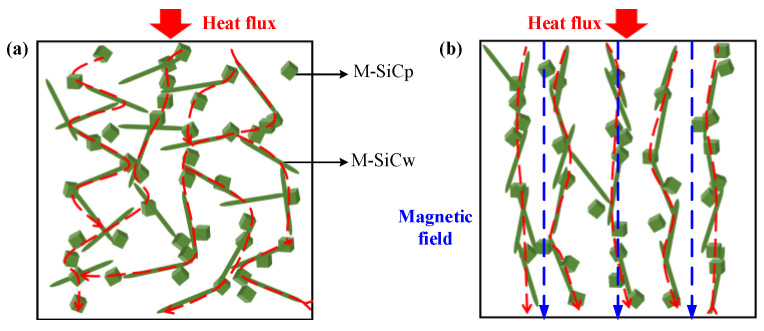
Schematic diagram of heat conduction inside M-SiC/EP composites. (**a**) Randomly oriented M-SiC/EP-N; (**b**) orderly oriented M-SiC/EP-C.

**Table 1 polymers-13-03028-t001:** The material parameters of SiC fillers.

SiC Filler	Morphology	Average Particle Size (μm)
SiCp	Particle	1.5
SiCw	Whisker	Length/diameter ≈ 30

**Table 2 polymers-13-03028-t002:** Types and abbreviations of SiC/EP composites.

Sample Category	Sample Abbreviation	Mass Fraction (wt%)		Action Mode of Magnetic Field
		SiCp	SiCw	
R-SiC/EP-N	R-SiC_p5_/EP-N	5	0	Without magnetization
	R-SiC_p5w0.1_/EP-N	5	0.1	
	R-SiC_p5w0.5_/EP-N	5	0.5	
	R-SiC_p5w0.9_/EP-N	5	0.9	
M-SiC/EP-N	M-SiC_p5_/EP-N	5	0	Without magnetization
	M-SiC_p5w0.1_/EP-N	5	0.1	
	M-SiC_p5w0.5_/EP-N	5	0.5	
	M-SiC_p5w0.9_/EP-N	5	0.9	
M-SiC/EP-P	M-SiC_p5_/EP-P	5	0	Pre-magnetization
	M-SiC_p5w0.1_/EP-P	5	0.1	
	M-SiC_p5w0.5_/EP-P	5	0.5	
	M-SiC_p5w0.9_/EP-P	5	0.9	
M-SiC/EP-C	M-SiC_p5_/EP-C	5	0	Curing magnetization
	M-SiC_p5w0.1_/EP-C	5	0.1	
	M-SiC_p5w0.5_/EP-C	5	0.5	
	M-SiC_p5w0.9_/EP-C	5	0.9	

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
