# Peer review of "Analysis of the Electrical and Thermal Properties for Magnetic Fe_3_O_4_-Coated SiC-Filled Epoxy Composites"

_polymers, 2021, doi:10.3390/polym13183028_

Round 1
Reviewer 1 Report
Comments for authors are in the attachment.

Reviewer 2 Report
Report of Manuscript polymers-13498777 for Polymers
Title: Analysis of the electrical and thermal properties for paramagnetic Fe3O4 coated SiC/ epoxy composites with different magnetization treatment by Jiale Wu et al.
This manuscript focus on the characterization of "epoxy composites" with different complementary techniques.
Overall Comments
-I think that this work could be of interest for the field of basic and applicative research of epoxy composites. The manuscript contains new information to justify publication. Different and complementary analyzes were carried out. The quantity and quality of the results presented is adequate with the standards of "polymers".
Issues
- The manuscript must be improved. It is poorly written; grammar and spelling should be revised. Several sentences are difficult to interpret because there are many repetitions. Many statements lack a bibliography.
- Some references are of little interest and can be improved.
- The results lack chemical-physical motivations. The paper is partially technically sound and it is limited and specific interest.
-The proposed goals by the authors are not clear. The logical thread of the manuscript is not clear. The results obtained are poorly presented. The future applications are lacking.
All these aspects need to be improved before any comments and considerations.
- The abstract could be rewritten and focuses on the results obtained.
- The conclusions need to be enriched to emphasize the applicability of the results found: this aspect is fundamental to the publication and impact of this manuscript.
I suggest the "Major-revision” encourage authors to resubmit a totally improved manuscript.
Reviewer 3 Report
Submitted manuscript is devoted to the study of the the electrical and thermal properties of Fe3O4 coated SiC/ epoxy composites. This is an interesting topic requested by the scientific society both from the point of view fundamental properties of composites and many different practical applications. Work has some potential and describes certain advances in the field of complex composites. However, it contains some flaws which make it non-acceptable even after a very major revision.
The process of the ordering of magnetic particles in an external magnetic field is quite well known and must be properly discussed and referenced (see examples: A. Hubert and R. Schäfer, Magnetic Domains. Berlin, Germany: Springer, 1998; Journal of Applied Physics 117, 123917 (2015); doi: 10.1063/1.4916700; Sensors 2018, 18(7), 2054; https://doi.org/10.3390/s18072054, etc.). Authors offer the title with words paramagnetic Fe3O4 and magnetization treatment without any estimation of the magnetic properties of initial components and different composites fabricated here. There is no evidence that they are releted to magnetite, which even in the most small size scale is not a paramagnet - Sensors 2020, 20(9), 2554; https://doi.org/10.3390/s20092554; Aphesteguy et al. Molecules 2014, 19, 8387-8401; doi:10.3390/molecules19068387; etc.), Ansari et al. Materials 2019, 12, 465; Yuvchenko, Tech. Phys. 2014, 59, 230–236; R. C. O’Handley, Modern Magnetic Materials: Principles and Application. New York: Wiley, 2000, p. 570.). VSM or SQUID magnetometry are necessary as well as EDX confirmation of the proposed models.
Table 2 has errors in description of significant numbers and not a single figure is given with error bars or experimental error estimation. For example, Fig. 12 is not valid without error estimation as all curves are very close.
Round 2
Reviewer 1 Report
Accept.
Author Response
Dear Reviewer,
Thank you very much for your time involved in reviewing the manuscript and your very encouraging comments on the merits.
Sincerely,
The Authors
Reviewer 2 Report
I am satisfied with the corrections done by the authors and consider that the article may be published in the present form.
Author Response
Dear Reviewer,
Thank you very much for your recognition on our work. We really appreciate your great comments on our work.
Sincerely,
The Authors
Reviewer 3 Report
Authors made an effort to improve the manuscript. However, part related to magnetism was not really addressed (neither suggestions nor analysis of the literature were taken into account). As it was in the first version
The process of the ordering of magnetic particles in an external magnetic field is quite well known and must be properly discussed and referenced (see examples: A. Hubert and R. Schäfer, Magnetic Domains. Berlin, Germany: Springer, 1998; Journal of Applied Physics 117, 123917 (2015); doi: 10.1063/1.4916700; Sensors 2018, 18(7), 2054; https://doi.org/10.3390/s18072054, etc.). Authors offer the title with words paramagnetic Fe3O4 and magnetization treatment without any estimation of the magnetic properties of initial components and different composites fabricated here. There is no evidence that they are releted to magnetite, which even in the most small size scale is not a paramagnet –Sensors 2020, 20(9), 2554; https://doi.org/10.3390/s20092554; Aphesteguy et al. Molecules 2014, 19, 8387-8401; doi:10.3390/molecules19068387; etc.), Ansari et al. Materials 2019, 12, 465; Yuvchenko, Tech. Phys. 2014, 59, 230–236; R. C. O’Handley, Modern Magnetic Materials: Principles and Application. New York: Wiley, 2000, p. 570.). VSM or SQUID magnetometry are necessary as well as EDX confirmation of the proposed models.
Round 3
Reviewer 3 Report
Futhors made special efforts and improved their work up to acceptable level.